# Avatar Intervention for Cannabis Use Disorder in Individuals with Severe Mental Disorders: A Pilot Study

**DOI:** 10.3390/jpm13050766

**Published:** 2023-04-29

**Authors:** Sabrina Giguère, Stéphane Potvin, Mélissa Beaudoin, Laura Dellazizzo, Charles-Édouard Giguère, Alexandra Furtos, Karine Gilbert, Kingsada Phraxayavong, Alexandre Dumais

**Affiliations:** 1Department of Psychiatry and Addictology, University of Montreal, Montreal, QC H3T 1J4, Canada; sabrina.giguere.2@umontreal.ca (S.G.); stephane.potvin@umontreal.ca (S.P.); melissa.beaudoin.1@umontreal.ca (M.B.); laura.dellazizzo@umontreal.ca (L.D.); 2Research Center of the University Institute in Mental Health of Montreal, Montreal, QC H1N 3V2, Canada; 3Faculty of Medicine and Health Sciences, McGill University, Montreal, QC H3G 2M1, Canada; 4Department of Chemistry, University of Montreal, Montreal, QC H2V 0B3, Canada; 5Services et Recherches Psychiatriques AD, Montreal, QC H1N 3V2, Canada; 6Institut National de Psychiatrie Légale Philippe-Pinel, Montreal, QC H1C 1H1, Canada

**Keywords:** cannabis use disorder, virtual reality therapy, psychotic disorder, mood disorder, addiction intervention, relational therapy

## Abstract

Cannabis use disorder (CUD) is a complex issue, even more so when it is comorbid with a severe mental disorder (SMD). Available interventions are at best slightly effective, and their effects are not maintained over time. Therefore, the integration of virtual reality (VR) may increase efficacy; however, it has not yet been investigated in the treatment of CUD. A novel approach, avatar intervention for CUD, uses existing therapeutic techniques from other recommended therapies (e.g., cognitive behavioral methods, motivational interviewing) and allows participants to practice them in real-time. During immersive sessions, participants are invited to interact with an avatar representing a significant person related to their drug use. This pilot clinical trial aimed to evaluate the short-term efficacity of avatar intervention for CUD on 19 participants with a dual diagnosis of SMD and CUD. Results showed a significant moderate reduction in the quantity of cannabis use (Cohen’s d = 0.611, *p* = 0.004), which was confirmed via urinary quantification of cannabis use. Overall, this unique intervention shows promising results. Longer-term results, as well as comparison with classical interventions in a larger sample, are warranted through a future single-blind randomized controlled trial.

## 1. Introduction

Over 4% of the global population has used cannabis in the previous year, making it the most-consumed drug in the world [1]. Particularly in North America, over 16% have used cannabis in the previous year based on data from 2017 and 2020 [2,3]. Prevalence has been increasing over the past decade, notably due to increased accessibility (e.g., legalization), isolation associated with COVID-19 lockdowns, and growing positive perceptions regarding its use [2,4,5,6]. Several factors can increase the likelihood of consuming cannabis, such as living with a severe mental disorder (SMD), including chronic psychotic (e.g., schizophrenia, schizoaffective disorder) and mood (e.g., bipolar disorder, major depression) disorders [7,8,9,10].

Multiple factors could explain this association. First, individuals from this population may have cravings of greater intensity due to their strong tendency to seek short-term rewards [11,12]. Second, there might be pre-existing common neurobiological vulnerability to SMD and substance dependence [13]. Lastly, widely spread beliefs about cannabis being harmless might prompt patients to use it in an attempt to temper some of their symptoms [14,15]. However, the literature tends to show the opposite. Notably, in the general population, cannabis use is associated with cognitive deficits and health-related issues [16,17,18]. In addition to these effects, cannabis use in individuals with SMD also leads to decreased medication compliance, higher rates of hospitalization, poorer quality of life, and higher risk of auto- and hetero-aggressive violence [19,20,21,22,23,24,25,26]. Studies have also shown that its use can lead to the exacerbation of psychiatric symptoms such as delusions and hallucinations, as well as precipitating or aggravating episodes of mania and depression [12,27,28,29]. Although the mechanisms are not fully understood, pharmacological and brain imaging studies suggest that psychotic and manic symptoms are partly due to dopaminergic hyperactivity. THC from cannabis binds to endocannabinoid receptors leading to a signaling cascade that decreases dopamine reuptake, which may promote dopaminergic activity and, consequently, exacerbate symptoms [27,30,31].

Despite all these negative consequences, up to 1 in 4 individuals with SMD will develop a CUD in their lifetime [8,10,32]. This problem is exacerbated by the fact that a CUD is more challenging to address in people with SMD than the general population [33,34]. That problematic raises major societal concerns regarding public health as well as direct and indirect costs (i.e., justice system, law enforcement, healthcare) [35,36]. In this sense, it is crucial to develop effective interventions to reduce cannabis use in individuals with a dual of CUD and SMD.

Psychosocial interventions are the first and only line of treatment of CUD since no pharmacological treatment has been approved for this use [37,38,39,40]. In the general population, motivational approaches and cognitive-behavioral therapy appear to have moderate effect sizes on cannabis use [41,42]. However, limited literature is available for those with a dual diagnosis of SMD and CUD. Based on the modest number of studies conducted on this population, no significant effects were observed for cognitive behavioral therapy, motivational interview or their combination [43,44,45,46,47,48]. Contingency management achieved a small but significant effect size regarding cessation of cannabis use, although it was not maintained over time [49,50]. Therefore, there is a crucial need for new interventional avenues in individuals with dual diagnosis.

Over the past few years, virtual reality (VR) has been integrated as a therapeutic tool in the treatment of various mental disorders (e.g., anxiety and psychotic disorders), with promising results [51,52,53,54,55]. This modality allows patients to be exposed to immersive emotion-inducing life contexts while supervised by a therapist [51,52,53,54,55,56]. It has been shown that these virtual experiences can reach an emotional intensity similar to that of everyday life, thereby allowing the transfer of learned strategies into the real-life [56]. Moreover, VR has the advantage that a scenario can be repeated as many times as necessary to achieve a specific goal. As for its use in substance use treatments, VR has been shown to have a greater ecological validity compared to traditional interventions [57]. Notably, VR has the potential to effectively induce cravings for nicotine, alcohol, cocaine, and cannabis [58,59,60,61,62]. Moreover, it has been proposed that repeated exposure to nicotine and alcohol in VR may lead to the extinction of the behavior [63,64,65]. Because of its versatility and capacity to recreate scenarios, VR has mostly been integrated in cue-exposure psychotherapeutic interventions [57]. These approaches aim to extinguish the conditioning associated with the substance using repeated exposure [66,67]. Thus far, the efficacy of such interventions has only been evaluated in nicotine use disorder. While therapies that only exposed patients to the substance did not yield significant results, VR interventions recreating more elaborate contexts achieved significant substance use reductions [63,68,69,70]. Therefore, learning appears to be best achieved when emotions and cravings are at their strongest, which could allow patients to transfer their new skills into their daily life [71,72]. Nevertheless, one crucial limitation common to all currently available VR interventions for substance use disorders (SUD) is that the interactions were limited (i.e., predefined situations and no dialogues in real time with avatar).

In order to integrate VR into the treatment of individuals with a dual diagnosis of CUD and SMD, our research team developed avatar intervention for CUD. This dialogical intervention could be achieved with ease as members of the research team already had the professional skills and the appropriate VR equipment to offer this type of intervention. Indeed, a treatment psychotherapeutic intervention using VR aiming to treat patients with treatment-resistant schizophrenia was developed by our team and was proven to have significant benefits for patients [73,74]. This novel intervention allows the personalization of the experience by recreating contexts eliciting emotions and cravings related to cannabis use in VR. To do so, avatars representing people related to each participant’s cannabis use (e.g., a person with whom they consume or a person generating emotions that precipitate cravings) are created and animated by the therapist. Therapeutic targets such as managing cravings and negative emotions, assertiveness, self-esteem, motivation for change, and relapse prevention are practiced in real time through dialogues between the participants and their avatars. This pilot clinical trial primarily aims to evaluate the efficacy of avatar intervention for the treatment of CUD in individuals with a comorbid SMD. Moreover, the secondary aims are to assess the acceptability and feasibility of the intervention as well as to estimate the amplitude of the observed effects to estimate the sample size needed in future, more extensive trials.

## 2. Materials and Methods

### 2.1. Participants

A total of 35 adult participants (32 outpatients and 3 inpatients) were recruited at psychiatric hospitals, the University Institute in Mental Health of Montreal and in the community. Of the 32 outpatients, the majority of them (*n* = 22) were referred by their clinical team and the rest referred themselves. All of them had a DSM-5 diagnosis of moderate or severe CUD as well as a DSM-5 diagnosis of a schizophrenia-spectrum disorder (i.e., schizophrenia, schizoaffective disorder, substance-induced psychotic disorder) and/or a mood disorder (i.e., bipolar disorder, major depression). The diagnoses were confirmed using the Structured Clinical Interview for DSM 5 [75]. For feasibility and follow-up purposes, participants were excluded if they had a neurological disease or an unstable serious physical illness. All participants provided informed written consent, and the trial was approved by the institutional ethics committee.

### 2.2. Design

This paper presents the results of a pilot clinical trial (Clinicaltrials.gov identifier: NCT03585127). Clinical evaluations were performed about one week before the beginning of the intervention (pre-intervention) as well as one week after the last interventional session (post-intervention). During the entire duration of the study, participants continued to receive their standard psychiatric care.

### 2.3. Avatar Intervention

Participants underwent at least 8 weekly sessions lasting between 60 and 90 min. The entire intervention was delivered by an experienced psychiatrist (AD) with 10 years of experience working with this complex population. The first session focused primarily on case formulation. The therapist started by drawing up a general portrait of the participant, including their consumption history, motivations, objectives, and expectations. The second session started with psychoeducation, notably by discussing the positive and negative effects of cannabis use. Then, the participant was invited to create avatar(s) in the VR environment representing people with a key role in their cannabis consumption (e.g., a friend, a drug dealer, or people who induce cravings and/or negative emotions). With the help of the therapist and a master's student (SG), participants could personalize their avatar’s physical appearance and voice (e.g., pitch). Finally, the remaining six sessions incorporated VR immersions during which participants could enter a dialogue with their avatar (animated by the therapist). The immersive sessions were separated into 3 phases, each lasting around 30 min:(1)Pre-immersion: Summary of the preceding week and determination of the objective(s) of this therapy session. The participant and the therapist decided together the aim of the scenario, the avatar that would be used for the dialogue and the VR environment (i.e., in a bar, an apartment, or a park).(2)Immersion: The participant sat in an adjacent room and was invited to enter a dialogue with their avatar in the VR environment. The therapist, who could see the participant through a one-way mirror, animated the avatar by having his voice modified in real-time as well as by controlling the avatar’s facial expression (i.e., angry, sad, joyful, fearful). During these interactions, participants were encouraged to practice their coping mechanisms and diverse skills.(3)Post-immersion: Debriefing of the participant’s experience, including the feelings that arose during the immersion.

Throughout the intervention, participants had the opportunity to work on these three main themes:(1)Relation to cannabis: Participants were invited to practice their self-affirmation, refusal techniques, and alternatives to consumption (e.g., new activities and hobbies).(2)Relationships with others: Participants worked on managing negative emotions (e.g., stress) and conflict resolution methods.(3)Relation to oneself: This intervention focused on the improvement of one’s self-esteem by targeting the participant's’ internal negative discourse, ambivalence, motivation for change, and perception of a future without cannabis use.

### 2.4. Set up Material

Participants underwent VR immersion facilitated by an Oculus Rift head-mounted display (Meta Quest, California, United States of America) The VR environment was created using a custom-made Unity 3D game engine, including unique avatars generated with the Morph3D Character System. While SALSA with Random Eyes Unity 3D extension was used to synchronize the avatar’s lips with its speech, a voice transformer (Roland AIRA VT-3) (Rolland Corporation, Ōsaka, Japon) was used to simulate a voice chosen by the participant in real time. Furthermore, the avatar’s facial expressions were programmed to reflect diverse emotions based on the Facial Action Coding system [76]. Prior to this study, this equipment was already being used to create realistic dialogues in a virtual context in order to treat individuals with treatment-resistant schizophrenia [73,74].

### 2.5. Clinical Assessments

Clinical assessments were administered before and after avatar intervention for CUD by a trained psychiatric nurse. The primary outcomes were the quantity and the frequency of cannabis consumption, both of which were assessed using the Timeline Follow-Back (TLFB), a self-reported questionnaire evaluating substance use in the previous week (e.g., cannabis, alcohol, cocaine). This tool has strong interrater reliability [77,78]. To ensure that cannabis had not been replaced by another substance, the use of alcohol and other recreational drugs was monitored. Therefore, a non-significant result would indicate that there is no increase in the consumption of other drugs. All quantities were converted into selling prices as reported by participants to have a comparable unit of measurement.

Secondary outcomes included the severity of problematic cannabis use, motivation for change, psychiatric symptoms, and quality of life. The severity of problematic cannabis use was assessed using the cannabis use problems identification test (CUPIT), a self-reported questionnaire with good to excellent test–retest reliability and internal consistency. This tool was chosen because of its excellent ability to discriminate severity subgroups longitudinally: non-problematic (score < 12), risky use (12–20), and problematic use (>20) [79]. Participants’ motivation to change their cannabis use habits was measured using the Marijuana Ladder, a tool with good concurrent and predictive validity assessing their readiness to change (pre-contemplation, contemplation, preparation, action, maintenance) [80]. Life satisfaction was assessed using the Heinrichs’ Quality of Life Scale (QLS), a 21-items tool covering different spheres of quality of life [81]. Symptoms of schizophrenia were evaluated using the Positive Additionally, Negative Syndrome Scale (PANSS). Evaluators were trained to administer the latter clinical scale by using a series of gold-standard videotapes and by conducting consensus ratings to ensure interrater reliability. This tool has been shown to have good interrater reliability, appropriate test–retest reliability, and high internal consistency [82,83].

Moreover, to confirm the self-report data on the amounts of cannabis consumed by the participants, urine levels of 11-nor-9-Carboxy-Δ9-tetrahydrocannabinol (THC-COOH) were quantified as tertiary outcomes [84]. A reversed-phase chromatography separation coupled to tandem mass spectrometry detection on a Xevo TQ triple quadrupole instrument (Billerica, MA, USA) was developed for THC-COOH quantification with THC-COOH d3 as an internal standard [84]. The overall method had a coefficient of variation lower than 4%, a quantification limit of 119 ng/mL, and a dynamic range of up to 700 ng/mL. All THC-COOH concentrations obtained via this approach were creatinine-normalized.

The intervention’s acceptability and feasibility were evaluated by collecting feedback from a subset of participants. To do so, semi-structured interviews were conducted using a series of questions derived from Feeley and Cossette’s research (2015) [85]. These questions were designed to obtain the participants’ opinions on various aspects of the intervention, such as the content, frequency, number and sequence of the sessions, adequacy as well as use of VR. These interviews aimed to gather valuable insights into the participants’ perspectives and experiences. Additionally, possible adverse events related to the intervention were monitored. All adverse events were presented to two independent team members to determine if each was attributable to the avatar intervention for CUD.

### 2.6. Analyses

Changes in reported outcomes during the assessment periods pre–post intervention were assessed using a linear mixed-effects model with maximum-likelihood estimations for missing data. Quantification of THC-COOH in urine was performed by applying the creatinine concentration ratio. Changes in THC-COOH were assessed using Wilcoxon for paired variables. In order to assess whether there was a patient profile associated with patients who completed the intervention versus those who dropped out, baseline sociodemographic and clinical characteristics as well as outcome at pre intervention were analyzed. For continuous variables, a *t*-test was performed for normally distributed data and a Wilcoxon–Mann–Whitney for non-normally distributed data. For dichotomous variables, a Pearson’s chi-squared test was used to assess categorical outcomes differences. In the case one cell or more in the contingency table had a frequency under 5, Fisher’s exact test was used as it is more applicable to smaller samples [86].

The statistical threshold for significance was set at *p* < 0.05. Effect sizes were categorized as small (0.2–0.5), medium (0.5–0.8), and large (>0.8) effects. All statistical analyses were performed using IBM’s Statistical Program for Social Sciences (SPSS) for Windows (Version 25, IBM, Armonk, NY, United States). 

## 3. Results

### 3.1. Sample Characteristics

Sample characteristics are found in Table 1. Overall, participants were mostly men (78.9%), white (89.5%), and unemployed (68.5%). The age ranged between 24 to 55 years old. Almost all participants had severe CUD (89.5%), and the remainder had a moderate CUD. The average age of cannabis initiation was 14.6 years old. More than one-third of participants had another SUD, notably alcohol (42.1%) and stimulants (e.g., cocaine, amphetamine; 52.6%). Most participants had a primary diagnosis of a schizophrenia-spectrum disorder (84.2%) as well as a personality disorder (antisocial or borderline; 64.7% of *n* = 17).

### 3.2. Intervention Efficacity

As presented in Table 2, several statistically significant improvements were observed between the baseline and the post-intervention evaluations. Notably, the quantity of cannabis use (number of grams/joints), assessed with the TLFB, moderately significantly decreased by 50% (d = 0.611, *p* = 0.004). The urinary THC-COOH quantification confirmed this reduction in 12 participants from whom urine samples were taken (*p* = 0.037). Additionally, there was a strong correlation between THC-COOH quantification and self-reported amounts of cannabis (r = 0.763, *p* = 0.01). A slight decreasing tendency was observed in the frequency of cannabis use, from 5.2 to 4.2 days per week (d = 0.313, *p* = 0.052). Moreover, cannabis use was not replaced by the use of another substance, as the amount of money spent on all other substances combined (alcohol and drugs, excluding cannabis) decreased from CAD 88.1 to CAD 17.3 on average (d = 0.397; *p* = 0.217). Regarding the severity of cannabis use measured with the CUPIT, there was a significant improvement following the intervention (d = 0.474, *p* = 0.046). Motivation for change assessed with the Marijuana Ladder went from an average stage of contemplation to preparation (d = 0.523, *p* = 0.046), and the quality of life seemed to be slightly increased but not significant (d = 0.220, *p* = 0.146). Therapy had no significant influence on psychotic symptoms, which remained stable.

### 3.3. Acceptability and Feasibility of Interventions

From the initial sample of 32 participants, 13 withdrew their participation during the intervention. From this number, seven decided to stop the therapy before the first immersive session. Reasons for withdrawal included lack of motivation and temporary cessation of the intervention during the COVID-19 pandemic. As the trial started, the dropout rate was noted to be very high (only 7 out of the first 18 participants completed the intervention; rate: 61%). In response to this, the recruitment methods were slightly changed in order to ensure the inclusion of participants motivated for intrinsic rather than extrinsic reasons (e.g., monetary gain, permission to go out despite the lockdown). Following these measures, a large decrease in the dropout rate was observed (12 of the last 14 participants completed the intervention; rate: 14%). The overall attrition rate was 38.7 (see Figure 1). There were no statistically significant sociodemographic or clinical differences between therapy completers (*n* = 19) and non-completers (*n* = 13). Regarding substance use profiles at baseline, severity of CUD, amount and frequency of cannabis use as well as motivation to change cannabis use were not significant. However, an amphetamine SUD was significantly associated with quitting (*p* = 0.003), while alcohol and cocaine SUDs were not. Finally, the type of referral, being by a clinical team or by self-referral, was not significant.

Regarding potential adverse events, it is of note that none of the participants who discontinued therapy has done so because of concerns regarding the adverse effects of therapy. During the duration of the intervention, two participants were hospitalized: one was determined to be attributable to factors unrelated to the intervention, and the other occurred before the first immersive session. Moreover, one participant visited a psychiatric emergency department. This event was partially related to the intervention, although many other factors came into play. Nevertheless, the participant ended up being stable enough to be discharged rapidly. Finally, one suicidal act and one case of suicidal ideation were reported to our team by the participants. However, these were not related to the intervention as acknowledged by the participants themselves and who are also known for a long-term history of episodes of chronic despair. Of the five participants who experienced these events, four completed the intervention once their psychiatric symptoms stabilized.

Semi-structured interviews were conducted until data saturation, i.e., 11 participants to inquire about their perspectives regarding the intervention. Overall, all of them found their intervention adequate in terms of content, frequency, number, and sequence of the sessions. Moreover, they were all satisfied regarding VR use (see Table 3). Additionally, participants reported that the intervention changed their perceptions (e.g., “Confrontation helped me a lot. It made me change my way of seeing and doing things”). Moreover, many of them also mentioned that the intervention helped them find new coping strategies (e.g., “It helped me to put things into practice, managing my environment in order not to consume”).

Among the 19 participants who completed the therapy, the average number of sessions was 10.2 (range: 8–15), of which 7.4 sessions included VR immersion. As this was a pilot project, the therapist remained flexible and adapted the number of sessions to each participant. That allowed our team to assess the minimal and maximal number of sessions required to meet therapeutic targets in a future larger clinical trial.

## 4. Discussion

In the context of the current rise in VR use to enhance conventional interventional approaches, the present pilot clinical trial evaluated a novel VR intervention to treat CUD in individuals with a dual diagnosis of SMD. This intervention integrates several pre-existing approaches (e.g., motivational approach, cognitive-behavioral, relapse prevention) while using VR as a therapeutic tool. Notably, VR allowed participants to work on their dysfunctional relationships leading to cannabis use (i.e., with cannabis, others, and self) in real-time. This intervention’s relational and immersive aspects are hypothesized to enable the transposition of learnings into everyday life [71], which would lead to significant improvements.

Results showed significant reductions in the quantity of cannabis consumed as well as in the severity of CUD. Moreover, an increase in the motivation for change and a trend towards a decrease in the frequency of use were observed. Results indicate that the participants have an easier time decreasing their number of joints per day rather than not consuming for an entire day, as the quantity dropped to a greater extent than frequency. In order to confirm this decrease in self-reported cannabis use, a cross-validation was performed using quantitative urinalysis as well as by ensuring that other substances did not replace cannabis. As observed in other studies using similar quantification methods, a strong correlation was observed between urinary and self-reported measures [87,88]. Furthermore, although the amount of money spent on other substances decreased by 75%, this difference was not statistically significant. Nevertheless, this result confirms that participants did not substitute their cannabis use for another substance. Thus far, most studies investigating different interventions in this comorbid population did not show reductions in cannabis use. The only exception was for contingency management, which showed a small but significant effect regarding cessation of cannabis use [49,50]. Lastly, a trend towards an improvement in quality of life was observed, though the result was not significant; psychiatric symptoms remained stable. However, it should be considered that it might take longer than one week after the end of the intervention for significant effects to be observed on participants’ quality of life and psychiatric symptoms. To the best of our knowledge, the impact of any intervention on these aspects has never been investigated in this population. However, a study conducted in the general adult population showed no difference in quality of life after 12 weeks of either decreasing or increasing cannabis consumption [89]. Regarding psychiatric symptoms, it can be hypothesized that decreasing cannabis use might improve them, as heavier consumption is associated with more severe symptoms [28].

Most importantly, this pilot clinical trial showed that the implementation of this intervention is both feasible and acceptable for participants. Findings related to the adverse events were similar to those observed when evaluating other psychosocial interventions [47]; however, adverse events were not reported in the vast majority of studies investigating the effects of interventions for CUD in this population. Given the complexity of this population and the unstable mental status of many participants, it was not surprising that there were episodes of disorganization during the duration of the study. To confirm that this would have happened regardless of the intervention, comparing with a group receiving no intervention (treatment as usual) or with another intervention would be relevant in a future study. Nevertheless, no incidents were primarily attributed to the immersion sessions.

Due to the validation of the motivation to undertake such an intervention at the time of recruitment, a drastic reduction in drop out was observed. These changes lead to a rate similar to those obtained in other studies involving this complex population [90]. It can also be hypothesized that participants did not drop out because of the new treatment modality (VR), as nearly 50% of them decided to stop the intervention before the first immersive session. These results emphasize the importance of questioning participants about the reasons motivating them to undergo such an intervention [91,92]. In our study, apart from an amphetamine SUD, it was not possible to identify which profile of participants would be likely to complete the intervention or not. A meta-analysis likewise identified SUD to stimulants as an element that influences the drop rate in a population with SMD [90]. Nevertheless, our results could not be generalized due to the small sample.

Although many interventions using VR have emerged in psychiatry, very few studies have been conducted to evaluate the associated cost–benefits [93]. Considering that cannabis use leads to increased hospitalization in our population and the large costs associated with each day of hospitalization, it may be cost-effective to use VR for the treatment of CUD in a population with SMD [94,95,96]. Additionally, technological advances in recent years have made VR equipment more affordable, which could eventually make it easily available and accessible in clinics, especially since therapists seem to be open to its use [56,97,98]. A future cost-effective study would confirm this.

Avatar intervention for CUD shows promising results in a small sample of individuals with a dual diagnosis of SMD. Nevertheless, this pilot clinical trial has some limitations deserving recognition. Firstly, this study did not include a control group. In order to justify the use of VR, it will be essential to conduct a randomized trial comparing avatar intervention for CUD to another classical intervention. Secondly, only short-term results were analyzed; however, participants who have completed the intervention are currently being followed-up for up to a year following the end of the intervention. Thus, a future analysis will be conducted to assess whether observed changes are maintained over time. Thirdly, the sample size was small, and the large attrition rate further reduced the number of participants. This might have led to a lack of statistical power; therefore, several associations might become significant by analyzing a larger sample. To address these issues, a larger single-blind randomized controlled trial comparing the avatar intervention for CUD to a traditional substance abuse intervention is underway (ClinicalTrials.gov Identifier: NCT05704582). Finally, the THC/CBD ratio was not considered, mainly because participants were generally unaware of what they were consuming. A potential solution to this problem would be to analyze a sample of each participant’s cannabis; however, this raises ethical concerns which might be complex to address.

In conclusion, this novel VR intervention was shown to induce moderate reductions in cannabis use in individuals with a dual diagnosis of SMD and CUD. Although this is a small pilot trial, this new avenue is promising since this population’s options are currently very limited. Considering the increase in cannabis use as well as in the THC content in cannabis over the last few years, it is crucial to develop effective interventions for CUD [99,100]. That is even more important for individuals with SMD, for whom cannabis use greatly impacts several spheres of their life (e.g., symptoms, functioning, quality of life). In order to validate the superiority of this intervention over the existing ones, as well as to determine whether it has an impact on symptoms and quality of life, a larger single-blind randomized control trial will be conducted.

## Figures and Tables

**Figure 1 jpm-13-00766-f001:**
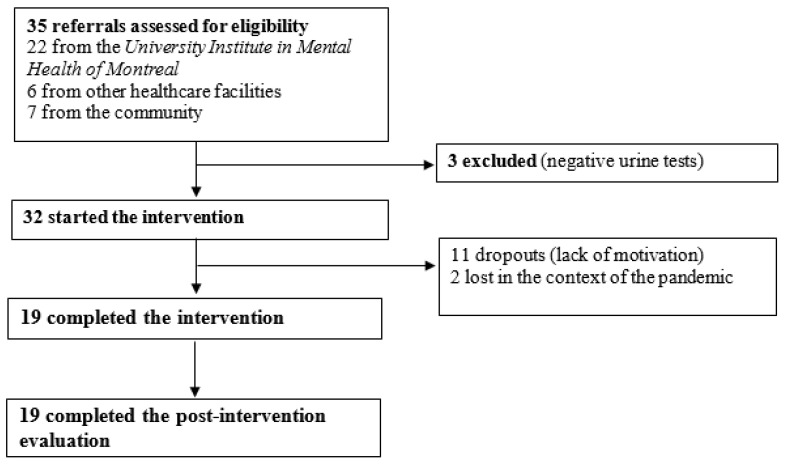
Trial profile of participants who underwent avatar intervention for CUD. There were 35 referrals provided by clinical teams and the community, 32 of whom were eligible.

**Table 1 jpm-13-00766-t001:** Baseline sociodemographic and clinical characteristics. (*n* = 19).

Caracteristics	Mean (SD) or n (%)
Age (years)	39.4 (9.86)
Sex	
Male	15 (78.9)
Female	4 (21.1)
Ethnicity	
White	17 (89.5)
Visible minorities	2 (10.5)
Current employ	6 (31.5)
Severity of cannabis use disorder	
Moderate	2 (10.5)
Severe	17 (89.5)
Age of onset of cannabis use	14.6 (1.58)
Other current substance use disorder	
Alcohol	8 (42.1)
Stimulants (cocaine, amphetamine)	10 (52.6)
Kétamine	1 (5.26)
Dual diagnosis	
Psychotic disorder	16 (84.2)
Bipolar disorder	1 (5.3)
Major depressive disorder	2 (10.5)
Personality disorder (*n* = 17) ^a^	
Antisocial	8 (42.1)
Borderline	6 (31.6)

SD: standard deviation; n: number of participants. ^a^ This measurement was added to baseline clinical evaluations later on during the trial; therefore, the first 13 participants were not assessed for personality disorders.

**Table 2 jpm-13-00766-t002:** Outcome at pre- and post-intervention.

	Pre-Intervention (*n* = 19)		Post-Intervention (*n* = 19)		Timepoint Comparisons	
	Mean	SD	Mean	SD	*p*-Value	Cohen’s d
TLFB cannabis quantity number of joints in last week	28.8	32.9	11.7	19.6	**0.004**	**0.611**
TLFB cannabis frequency days/in last week	5.16	2.76	4.21	3.24	0.052	0.313
CUPIT	38.5	10.7	33.0	12.4	**0.021**	**0.474**
Marijuana Ladder	6.5	2.23	7.70	2.22	**0.046**	**0.523**
QLS	60.8	19.4	64.8	17.6	0.146	0.220
PANSS	70.1	11.4	67.1	16.1	0.286	0.214
Money spent on drugs and alcohol (CAD per week) Of note: cannabis use was excluded	88.1	252	17.3	33.6	0.217	0.397

Linear mixed models with maximum-likelihood estimation were used to estimate *p*-values and effect sizes. Statistically significant differences (*p*-value < 0.05) are in bold. SD: standard deviation; TLFB: timeline follow-back, CUPIT: cannabis use problem identification test, QLS: quality of life scale, PANSS: positive and negative symptoms scale.

**Table 3 jpm-13-00766-t003:** Participants’ satisfaction with each component of the intervention. The score varies from 0 (unsatisfied) and 5 (very satisfied) (*n* = 11).

Components	Mean (SD)
Content of the sessions	4.45 (0.69)
Frequency of the sessions	4.45 (0.69)
Number of sessions	4.27 (0.79)
Adequacy	4.72 (0.65)
Sequence of the sessions	4.72 (0.65)
Use of virtual reality	4.45 (1.04)

SD: standard deviation.

## Data Availability

The data presented in this study are available upon reasonable request to the corresponding author.

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
