# Peer review of "Avatar Intervention for Cannabis Use Disorder in Individuals with Severe Mental Disorders: A Pilot Study"

_jpm, 2023, doi:10.3390/jpm13050766_

Round 1
Reviewer 1 Report
THis is an important study about psychotherapy of cannabis use disorder in severe mental disorders. The formal aspects and the English language are very well done. I would like the authors to mention which mental disorders can be aggravated by the cannabis use and why.
Reviewer 2 Report
This is an interesting pilot study of an intervention for cannabis use disorder comorbid with severe mental disorder. The main problem that I see in this study is the lack of a control group. We don’t know if this kind of intervention is better than other, or usual treatment.
Other comments:
-
- Line 37: “including” can be omitted
- The explanation of the avatar intervention in the introduction could be shortened, because it is extensively explained in the methods section.
- It is not clear to me how where the participants recruited. Which kind of institution is the University Institute in Mental Health of Montreal? Were the patients admitted for an episode, or is it a residential setting?
- Results: trends are not significant results, this should be considered (quality of life, decrease of consumption). Regarding the amount of spent money in drugs, you report a p of 0.178, is this correct? Trends and not significant results should not be reported as significant. Although you mention it, it seems that you consider some results as significant (also in the discussion)
- Attrition rates are high in this study. Did you compare the clinical characteristics of those patients who continued and who abandoned the study?
I Maybe your conclusions are too enthusiatic, considering the limitations. You should remark the limitations and the preparation of a new study with a control group. This work that you are presenting is an exploratory one, and should be discussed considering its limitations.
- In summary, the attrition rates and the selection of patients suggest that this intervention could be useful only for a reduced group of patients, with very specific characteristics. This could have cost-effective implications for this kind of interventions. This should be commented in the limitations.
Reviewer 3 Report
From my point of view, the present study is very well prepared and very well presents a new topic that has the potential for development in the future. Although the use of virtual reality in healthcare is not new, the use of Avatar as a form of therapy within a customized version is a very original idea.
The study is supported by extensive and relevant literature, I only recommend to consider the inclusion of source 3 in the introduction (the source is from 2018, presenting data from 2017, however the authors refer to data from last year - i.e. 2022), this is the only minor thing I recommend to modify.
The authors explain the design of the study very well. Personally, I find the control group based design lacking, however, the authors themselves acknowledge this limitation with the promise of a follow-up study. At the same time, the sample of patients is quite small after some dropped out of the study, but this limit is also acknowledged and the authors do not go into extensive generalizations, so this is also fine.
The authors write about three phases of immersive sessions (pre-immersion, immersion, post-immersion) - personally, I would be interested to know how much time each sub-phase took, or if this cannot be determined, what percentage of the total time stated was taken up by each of the sessions.
Only one of the interviews conducted is quoted in the text, personally I would not be opposed to more, but this aspect does not detract from the quality of the text.
